# Assessment of Cracking in Masonry Structures Based on the Breakage of Ordinary Silica-Core Silica-Clad Optical Fibers

Sergei Khotiaintsev [1,*] and Volodymyr Timofeyev [2]

1   Faculty of Engineering, Universidad Nacional Autonoma de Mexico, Coyoacan, Mexico City 04510, Mexico
2   Faculty of Electronics, National Technical University of Ukraine "Igor Sikorsky Kyiv Polytechnic Institute", Peremohy Street 37, 03056 Kyiv, Ukraine; v.timofeyev@kpi.ua
*   Correspondence: sergeikh@unam.mx; Tel.: +52-55-5468-5757

**Abstract:** This paper presents a study on the suitability and accuracy of detecting structural cracks in brick masonry by exploiting the breakage of ordinary silica optical fibers bonded to its surface with an epoxy adhesive. The deformations and cracking of the masonry specimen, and the behavior of pilot optical signals transmitted through the fibers upon loading of the test specimen were observed. For the first time, reliable detection of structural cracks with a given minimum value was achieved, despite the random nature of the ultimate strength of the optical fibers. This was achieved using arrays of several optical fibers placed on the structural element. The detection of such cracks allows the degree of structural danger of buildings affected by earthquake or other destructive phenomena to be determined. The implementation of this technique is simple and cost effective. For this reason, it may have a broad application in permanent damage-detection systems in buildings in seismic zones. It may also find application in automatic systems for the detection of structural damage to the load-bearing elements of land vehicles, aircraft, and ships.

**Keywords:** structural health monitoring; seismic assessment; detection of structural cracks; masonry buildings; optical fiber; distributed optical fiber sensors

## 1. Introduction

*1.1. Earthquakes: A Need for Rapid Assessment of Structural Damage*

Earthquakes destroy buildings and cause the loss of human lives in various regions of the world. The crisis situations that follow have many complex aspects [1]. Among them, a quick assessment of damage to buildings and the scope of destruction in the affected region is crucial to making the right decisions. Thus, it is better to have rapidly available, "good enough" analysis, rather than "perfect" information and analysis that comes too late. Late analysis, no matter how good, is of little use in designing immediate life-saving interventions" [2].

Modern buildings are usually quite resistant to damaging factors, but some still suffer from strong earthquakes, the degradation of materials, land sinking, etc. [3,4]. Moreover, many old buildings are exposed to great risks from earthquakes and other damaging factors [5,6]. The rapid detection of partial, frequently non-visible damage to buildings is particularly important since the degree of risk that it poses is high. However, such a risk is not always visible to the occupants of the buildings or to rescue teams. Moreover, timely information allows decision-makers to take appropriate measures, and thus avoid more serious consequences. Permanent seismic structural health monitoring systems can significantly reduce the human and economic losses from earthquakes. Given the millions of public and residential buildings, and the common dwellings in seismic zones, installing permanent structural health-monitoring systems on such buildings is feasible, provided that such systems are simple and cost-effective [7].

This paper presents studies on the suitability and accuracy of detecting structural cracks in brick masonry by exploiting the breakage of ordinary silica optical fibers bonded

to its surface. Brick masonry is common in buildings in many locations where baked clay brick is the main building material [8,9]. Furthermore, virtually the same technique can be applied to other masonry structures found worldwide [10,11].

### 1.2. Methods for Detecting Damage to Civil Engineering Structures

There is a variety of visual, mechanical, electrical, acoustical, computer vision, global dynamic behavior, and other methods that are used for damage detection in civil engineering structures [12–21]. Nevertheless, many are susceptible to the adverse effects of moisture, chemical corrosion, electromagnetic interference, and lightning discharges. These factors make it rather difficult to use these tools on large structures, especially outdoors.

Over the past few decades, optical fiber sensors have reached a high level of sophistication and performance, and they are increasingly used for structural health monitoring (SHM). The advantages of using optical fiber sensors in civil engineering are numerous. Firstly, the optical fibers (shown schematically in Figure 1) are made of dielectric materials, and, therefore, they do not conduct electricity and are immune to electromagnetic interference. Secondly, they are chemically inert, and therefore immune to adverse climatic factors.

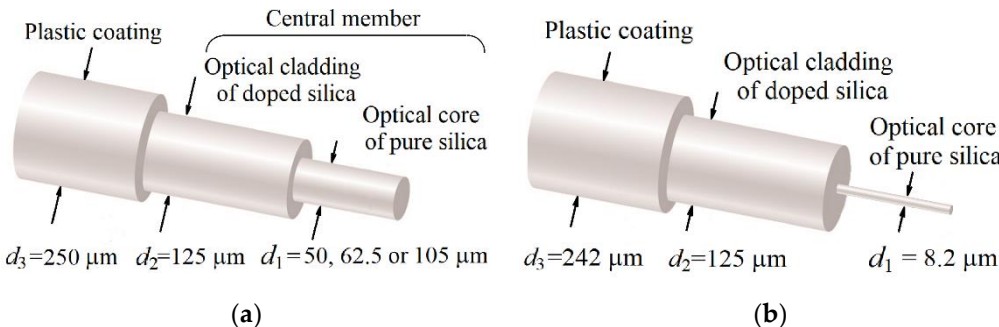

**Figure 1.** Schematic drawings of ordinary communication-grade silica-core silica-clad optical fibers: (**a**) multimode; (**b**) single mode.

The optical-fiber sensor systems that are most commonly used for SHM employ Optical fiber Bragg gratings (FBGs), optical frequency domain reflectometry (OFDR), Brillouin scattering optical time-domain analysis (BOTDA), and optical fiber Fabry–Perot resonators (FPRs). These optical-fiber sensor systems are highly sensitive to strain and have a distributed sensing capability (for details, see the reviews [22–28] and references therein).

Monitoring strain in structural elements is a useful way to confirm and ensure that bridges, buildings, etc., are indeed behaving as expected. Even in cases of seemingly proper structural behavior, these sensors provide useful information about the degree of coincidence between the predicted and observed behavior of the structure. Moreover, monitoring strain in existing buildings makes it possible to ensure their safety during nearby construction work or other hazardous activities. In addition, monitoring strain in buildings threatened by environmental or man-made factors helps specialists to take timely remedial measures. Some buildings from the past centuries are already equipped with distributed optical fiber sensor systems that provide useful information about the structural behavior of the buildings [29–33].

Moreover, distributed optical fiber sensors provide indirect information about structural damage. Anomalous strain concentrations in the vicinity of cracks indicate the cracking of structural elements [34–40]. This allows specialists to locate cracks and estimate their width using strain measurements and indirect numerical methods developed for this purpose. Such estimations have been carried out on reinforced concrete (RC) beams [34–38]. However, the maximum width of the cracks evaluated using this technique did not exceed 0.4 mm. In a previous study, structural cracks were detected in masonry, but the relationship between an anomalous strain and the respective crack width was not established [40].

Despite the impressive achievements in assessing the strain and temperature distribution in structures with the help of FBG, BOTDA, OFDR, and FPR-based sensors, the drawbacks of these sensors include the high cost of optical signal analyzers (also called interrogators) and the FBG, OFDR, and FPR gauges (the estimated cost of a high-resolution OTDR is USD 30,000–40,000; the cost of an FBG interrogator is USD 10,000–50,000; the cost of an OFDR interrogator is USD 60,000–100,000; and the cost of a high-performance BOTDA analyzer reaches USD 100,000, with the cost being highly dependent on the specific configuration and technical specifications. In addition, FBG and OFDR sensor systems require special optical fibers with embedded Bragg gratings, which have an estimated cost of USD 500–1000 per 100 m), as well as the need for specially trained personnel to work with these systems and interpret the results of the strain measurements.

Given the high cost and complexity of the said optical fiber sensor systems, FBG, BOTDA, OFDR, and FPR-based sensor systems are mainly used to monitor new original structures and precarious buildings to ensure that they behave properly and safely in real-world conditions [41,42]. For mass application in permanent damage-detection systems, which this work targets, only a much simpler and cheaper method of detecting structural damage is suitable.

### 1.3. Cracking of Load-Bearing Structural Elements

The cracking of the load-bearing structural elements of buildings is a significant indication of structural damage. The degree of danger of a crack is influenced by many factors—the type of building, its structural features, and other factors. Generally, the wider the crack, the greater the structural risk that it poses. There is no generally accepted classification of cracks in buildings. Various manuals and handbooks treat the danger posed by cracks differently [43–45]. In most documents, microcracks and cracks less than 1 mm wide are considered insignificant. According to the manual in [43], cracks from 1 mm to 3 mm wide in masonry present a medium structural risk, while in the case of wider cracks, the structural risk is high. The handbook in [44] states that cracks wider than 3 mm in concrete and masonry pose a serious structural risk and require a detailed structural evaluation. This is in contrast with the classification in [45], which considers that only a few 3-millimeter-wide cracks or a single crack over 5 mm wide in masonry buildings present "serviceability" issues. A stringent crack-width limit provides more safety. For this reason, this work aimed to detect cracks with a width of 1 mm or more, as they are capable of posing a structural hazard in masonry after earthquakes and causing other damaging effects.

### 1.4. Optical Fiber Breakage as an Indication of Cracking of Structural Elements

The appearance of structural cracks can be indicated by the breakage of optical fibers that are attached to the element or embedded in it and cross the crack. A fiber break causes a sharp decrease in the intensity of optical radiation passing through the fiber, which is easily detected by appropriate instruments. This method was proposed and applied to detect cracks in steel and aluminum components, as well as in composite materials, as early as the 1980s [46,47]. However, it was believed that this method produced poor results in detecting structural damage [48]. This opinion may be the result of the following:

- First, the optical coupling between the two parts of a broken fiber depends not only on their axial displacement but also on many random factors, including the angle between the crack and the fiber axis, the shear displacement of the two parts, and the roughness of their end faces. Therefore, signal loss cannot be uniquely related to the expansion of the crack.
- Second, optical fibers, as a brittle material, break at slightly different structural crack widths each time (this property is discussed in detail in the following sections).
- Third, the pioneering work did not demonstrate a practical cost-effective implementation of this method. These might be the three main reasons as to why it has never been used to detect structural damage in buildings.

### 1.5. Ultimate Strength of Optical Fibers

The ultimate strength is the maximum stress that a material can withstand before rupture. The ultimate tensile strength of optical fibers varies among supposedly identical specimens. As with any brittle material, the ultimate tensile strength of optical fibers of silica depends on the size and distribution of flaws, which are random [49,50]. Therefore, a breakage of an optical fiber cannot be attributed to a single strain or a single structural crack opening. Instead, a probabilistic rather than a deterministic definition describes the width of a structural crack when the crack is detected through a breakage of an optical fiber [49,51].

It should be noted that silica optical fibers are subjected to a tensile proof test at a specific stress of $\sigma_{pt}$ = 0.69 GPa (100 kpsi) at production facilities. That is, the silica optical fibers available on the market are guaranteed to withstand a stress of $\sigma_{pt}$ = 0.69 GPa. This stress corresponds to a tensile force of $F$ = 8.6 N in the case of a silica-core silica-clad optical fiber with a cladding diameter of $d_2$ = 125 um. Furthermore, the ultimate tensile strain, $\varepsilon_u$, of thin silica optical fibers is much higher than that of brick masonry, with their values being about 0.035 and 0.0001, respectively [52,53]. That is, masonry cracks are under a much lower strain than the silica optical fibers bonded to the masonry. Therefore, the optical fiber bonded to masonry initially stretches under masonry deformation and the onset of small structural cracks. It then breaks at a certain crack width, $\delta_c$. A particular crack width, $\delta_c$, depends on the ultimate strain of the optical fiber, $\varepsilon_u$; the bond between the silica cladding and the plastic coating; the coating material and its stiffness; the bond between the coating and the binder; and the stiffness of the binder [54–58].

### 1.6. The Authors' Previous Research

The authors previously investigated the possibility of detecting cracks in RC beams using fiber breaks [59–61]. Several embedding techniques were tested and twelve types of optical fibers were studied both in terms of their survival rate and their abilities to detect cracks. Large-diameter optical fibers successfully detected the progressive cracking of the RC beams under increasing load up to the ultimate failure of the beams. However, the correlation between fiber breakage and crack width could not be established due to the impossibility of assessing the width of internal cracks in a concrete beam.

Moreover, the prospects of detecting structural cracks in brick masonry using the present method were investigated in [62]. In this work, optical fibers were bonded to separate baked clay bricks that were pressed tightly together and laid on a platform. Then, the bricks were pushed apart at a strain rate of 0.25/min until the optical fiber broke [63]. Optical fibers of silica core, silica cladding, various coating materials, and various adhesives were tested. The most rigid bonding was achieved in the case of acrylate-coated optical fibers bonded to bricks with an epoxy adhesive. Further progress was described in the conference publication [64], which reported the detection of cracks with a minimum width of 1 mm in masonry using optical fibers.

### 1.7. Objectives

The general objective of this work was to evaluate the suitability and accuracy of detecting structural cracks in brick masonry by exploiting the breakage of ordinary silica optical fibers bonded to its surface. Of particular interest were the behavior of optical signals upon the deformation and cracking of the test specimen and the relationship between fiber breakage and the crack width that caused the breakage.

The specific objectives included the choice of the most suitable optical fiber for this application, the fabrication of a brick masonry test specimen, the instrumentation of the specimen with an array of optical fibers, the design and implementation of the monitoring system, and the load testing of the sample. Moreover, the specific objectives included analyses of the results: deformations of the specimen under load, the behavior of the optical signals under the deformation and cracking of the test specimen, and a statistical and probabilistic evaluation of the results.

## 2. Materials and Methods

### 2.1. Choosing the Right Optical Fiber

There are a variety of silica-core silica-clad optical fibers of different core and cladding diameters available on the market. The silica core and cladding constitute a monolithic silica member. The tensile strength of a silica-core silica-clad optical fiber depends on the overall diameter of the silica member, $d_2$, as well as on the ultimate stress of the fiber, $\sigma_u$. In practice, the tensile strength of 20 m fiber samples averages about 4 GPa [65]. The average ultimate strength, $\sigma_u$, increases with a decrease in the length of fiber samples, since the probability of a flaw in a sample is inversely proportional to its length.

However, the type of coating material affects the strength of optical-fiber bonding to a structural element. Based on the above considerations and previous research [61], and with the aim of detecting structural cracks in the mm range [43,44], acrylic-coated optical fibers of the smallest available cladding diameter $d_2$ = 125 um were chosen for this study. Among the fibers in this group, the optical fibers with the largest available core diameter $d_1$ = 105 um were chosen and used (a large core diameter facilitates the efficacious coupling of inexpensive light sources—light-emitting diodes (LEDs) and optical fibers). The optical fibers chosen had an acrylate coating with a diameter of $d_3$ = 250 um, numerical aperture of $NA$ = 0.22, and optical attenuation of $\alpha$ = 15 dB/km @ $\lambda$ = 633 nm.

The suitability of these optical fibers for the present application was verified by a simple tensile test, similar to that in [62], on 20 samples of these optical fibers. The test consisted of bonding an optical fiber sample to two separate ceramic bricks and pushing the two bricks apart at a strain rate of 0.25/min until the breakage of the fiber [63]. This led to a steep decrease in the pilot optical signal transmitted through the fiber to almost zero. The test yielded a minimum detected crack of about 1 mm wide, a mean structural crack width of $M$ = 2.34 mm, a variance of $\sigma^2$ = 0.79 mm$^2$, and a standard deviation of $\sigma$ = 0.89 mm (see Figure 2). (The zero probability of failure at a structural crack width range of $0 < \delta_c$ 1 mm is due to the screening out of optical fibers that failed the proof tests of $\sigma_{pt}$ = 0.7 GPa at the production facility.) These cracks fitted a range of 1 mm to 5 mm wide, which corresponds to a medium-to-high structural hazard for masonry [43,44]. Therefore, the above-named optical fiber was used in this work.

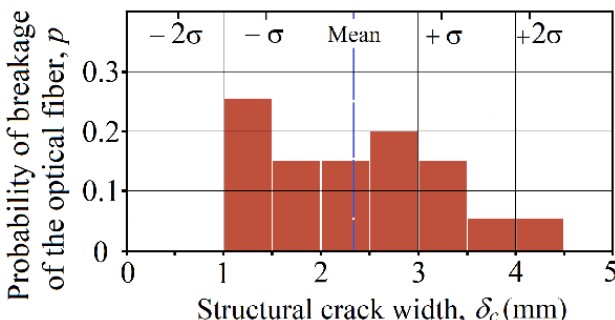

**Figure 2.** Probability of breakage of the optical fiber, $p$, as a function of structural crack width that causes the crack, $\delta_c$, obtained by performing a simple tensile test and determined for sequential intervals of a width of 0.5 mm.

### 2.2. Test Specimen

A clay brick masonry specimen was designed, fabricated, instrumented with the optical fibers, and subjected to a vertical load of the magnitude encountered in earthquakes. It was a stack with dimensions of 520 (W) × 520 (D) × 510 (H) mm$^3$ of baked clay bricks of 60 × 120 × 240 mm$^3$ and a sand–cement mortar (proportion 2:1, type II Portland cement and fine pumice sand). All the aforesaid components are typical of the clay brick masonry elements of dwellings and other brick masonry structures in central Mexico. To achieve a uniform compression load distribution on the brick stack, two RC platens, each about 100 mm thick, were built on the top and bottom of the stack. There were five reinforcing

steel bars of diameter $3/4''$ in each platen. The total height of the stack with two platens was 710 mm. The stack is shown in Figure 3a–c.

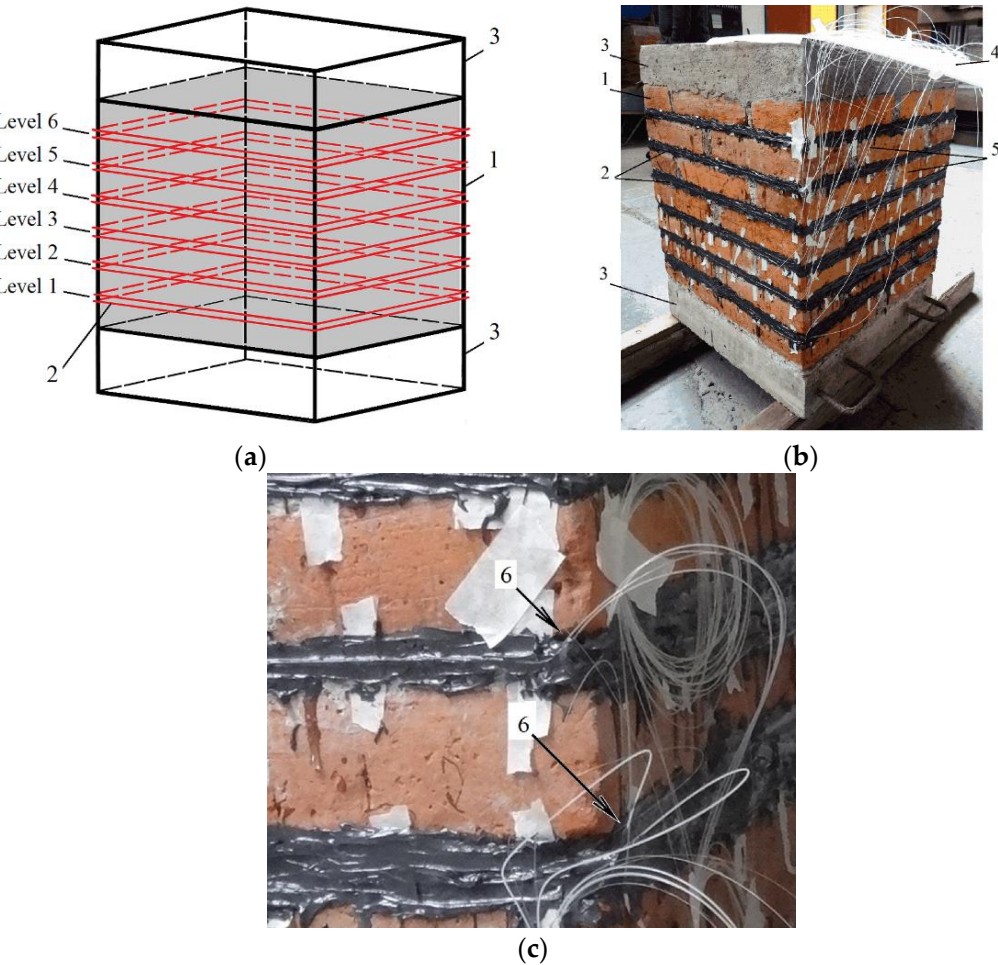

**Figure 3.** Brick masonry specimen instrumented with the optical fibers: (**a**) schematic; (**b**) photograph; (**c**) an enlarged view of the points of connection of optical fibers to the specimen (photos). (1) Brick masonry specimen; (2) segments of optical fibers bonded to masonry that serve as crack detectors (not visible under opaque epoxy adhesive); (3) RC platen; (4) tray with free ends of the optical fibers; (5) free ends of the optical fibers; (6) points of connection of the optical fibers to the masonry specimen.

The mortar of the brick specimen was cured for 28 days. Then the specimen was instrumented with the optical fibers, transported to the load machine, and installed in its test frame, as shown in Figure 4a. Moreover, the specimen was instrumented with eight electrical linear variable displacement transducers (LVDTs), in compliance with compression test methodology generally accepted in structural mechanics. The LVDTs were installed on the four faces of the brick specimen in the horizontal and vertical directions, as shown in Figure 4b,c. The transducers measured the horizontal and vertical deformations of the specimen in the central part of each lateral face at a length of 250 mm.

*2.3. Optical Fiber Array*

Segments of optical fibers, each about 5 m long, were prepared in advance. The ends of the optical fibers were cut at a right angle, and the quality of the cut and the integrity of the fibers were checked. The optical fibers were placed on a special tray that protected the fibers from possible damage, and they were delivered to the test site. The specimen was instrumented with twelve optical fibers. The optical fibers were bonded to the specimen over its perimeter at several (six) equidistant horizontal planes. A horizontal position for the optical fibers was chosen because cracks in masonry stacks subjected to a compression

load arise predominantly in the vertical direction. Two optical fibers were bonded near each other in each of the six planes of the specimen at about a 70 mm distance, as shown schematically in Figure 3a, using a two-component industrial-grade epoxy adhesive.

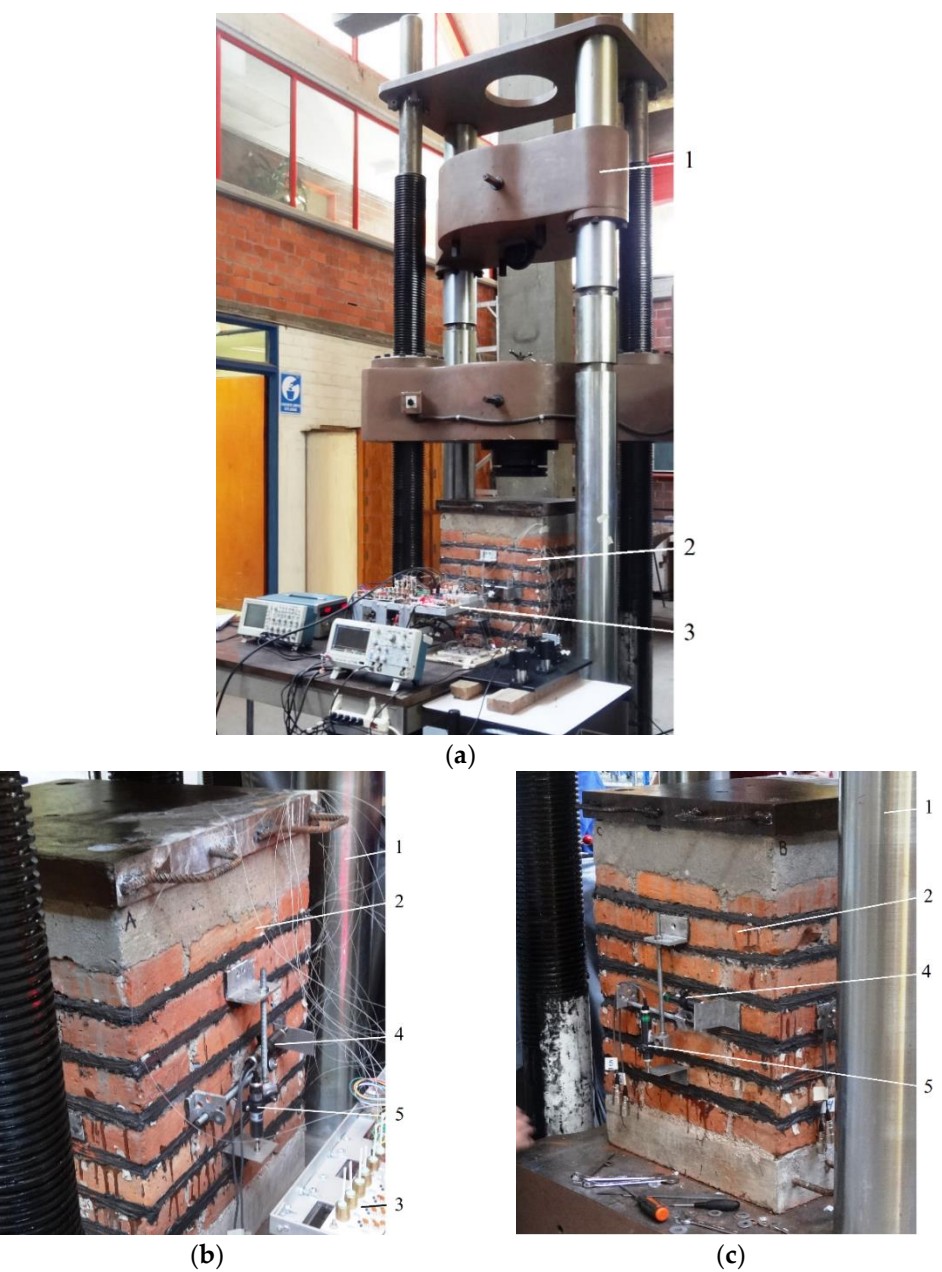

**Figure 4.** Brick masonry specimen: (**a**) instrumented with the optical fibers and LVDTs and installed in the test frame of the load machine; (**b**) the specimen front face (A); (**c**) the specimen back face (C). (1) Universal load machine; (2) brick masonry specimen; (3) opto-electronic monitoring system; (4) horizontal LVDT; (5) vertical LVDT.

The optical transmission of all 12 fibers was verified after bonding and again after 24 h. The aim was to identify possible damage to the optical fibers that could occur during bonding to the brick specimen or could be caused by the volumetric shrinkage of the adhesive due to its polymerization. The check was carried out with a common LED lamp. Each optical fiber was illuminated at one end, and the light coming out of the other end was observed. The inspection showed that all the fibers transmitted light normally; that is, none of the fibers were damaged during the process of bonding and polymerization shrinkage.

## 2.4. Opto-Electronic Monitoring System

To monitor the integrity and breakage of the optical fibers of the crack detection array, a special optoelectronic system was used, the configuration of which is schematically shown in Figure 5.

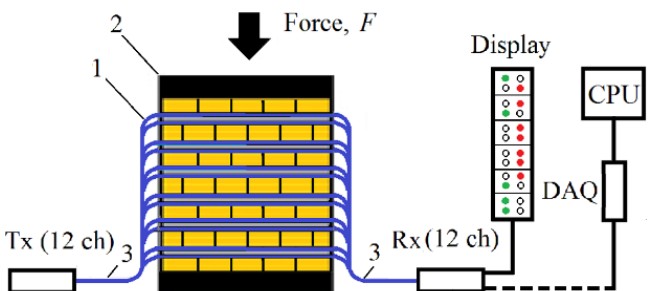

**Figure 5.** Schematic diagram of the opto-electronic monitoring system: (1) optical fibers; (2) test specimen; (3) array of 12 optical fibers for detection of structural cracks; Tx—optical transmitter, 12-channel; Rx—photoreceiver, 12-channel; DAQ—data acquisition module (multichannel analog-to-digital converter); CPU—computer central processing unit for storing the measured optical transmission in digital form; Display—an indicator of the structural damage in visual form.

The system monitored the intensity of the optical radiation transmitted through all optical fibers of the array by the multichannel optical transmitter, Tx. The breakage of the optical fiber led to a decrease in the signal intensity. The signals of all optical fibers were received by the multichannel optical receiver, Rx, and converted from analog to binary (two-level) form, that is, "1" or "0", by comparing the signal values against a given value. The zero level was interpreted as a structural crack, and the state of the corresponding color indicator on the display changed from green to red.

The optical radiation generated by the transmitter Rx was a continuous 1 kHz sine optical pilot signal. The light sources were LEDs operating at a wavelength of 633 nm (visible red light). Each LED was coupled to one of the optical fibers of the array. The outputs of the optical fibers were connected to the respective photoreceivers. Each measuring channel had a threshold amplitude device, the purpose of which was to automatically signal a fiber break. This device turned on a light indicator on a special display when the signal dropped below a certain level. The threshold was set at 50% of the nominal, as it was assumed that such a threshold would minimize the false alarms of the signaling system caused by possible fluctuations in light sources and all kinds of noise.

For this experiment only, the monitoring system was complemented with a 16-channel data acquisition device (DAQ) and a CPU that stored the data of all measuring channels in digital form. It should be noted that neither a DAQ nor a CPU is needed for a practical crack detection system. Instead, a practical system would require a means of transmitting alarms to the system users. (The cost of all elements and components of this monitoring system is about USD 300. The wholesale cost of the multimode optical fiber is about USD 50 per 1 km.).

## 2.5. Structural Load Test

The compression load test was carried out by means of a 200 kN universal load machine. The test was carried out in several stages of loading and unloading, until the specimen began to split off (at $F = 1334$ kN). The vertical and horizontal deformations of the specimen were measured by the electrical LVDTs installed in the central part of each lateral face of the specimen over a length of 250 mm.

The optical signals of all fibers were measured continuously by the above monitoring system, stored in the CPU, and post-processed after finalizing the test. The breakage of the optical fibers was easy to observe visually due to the red light emanating from the broken optical fibers. Photographs were taken of the locations where the fiber breakage

occurred. The width of the structural cracks causing the fiber breakage was determined by measuring the width of the cracks in the photographs. With this method, the actual width of the cracks may be somewhat smaller than that measured, since the cracks may expand from the time the fiber breaks to the time the photo is taken. Nevertheless, the use of other, more accurate methods to measure the width of structural cracks at the moment of fiber breakage requires a much more complex and expensive infrastructure, and was therefore not possible in this experiment.

## 3. Results

### 3.1. Data Overview

The changes in the load and deformation of the specimen, measured by the electrical LVDTs in the central part of each face of the specimen over a length of 250 mm, are represented by graphs in Figure 6. Some differences in the horizontal and vertical deformations were observed on the four lateral faces of the specimen. These can be explained by the heterogeneity of the materials, the geometric asymmetry of the specimen, and eccentric loading. It is difficult to avoid these factors in these types of tests.

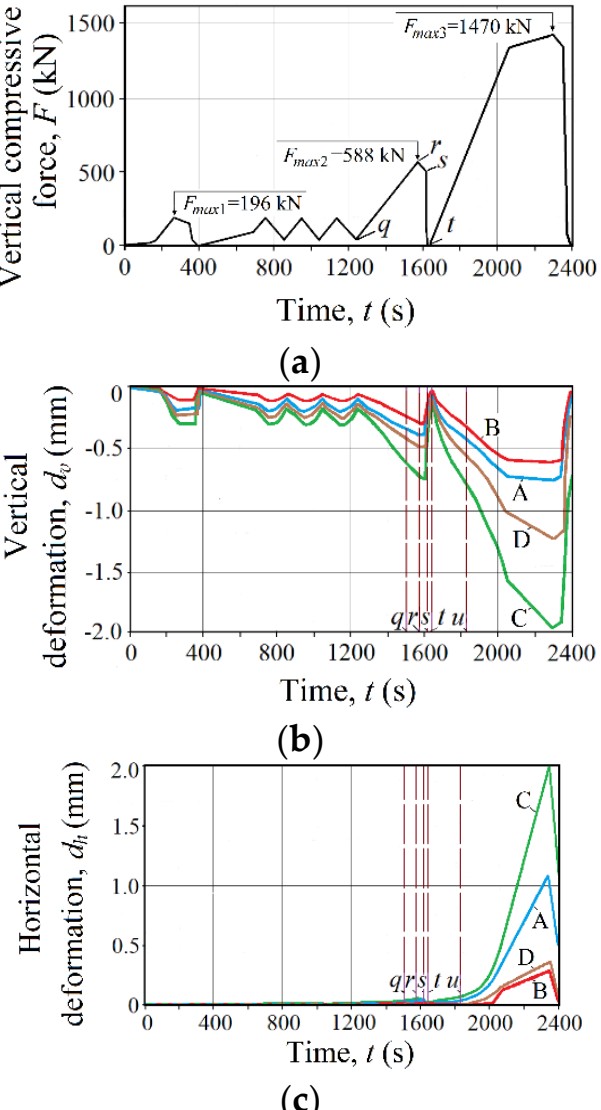

**Figure 6.** Specimen behavior under load: (**a**) variation in compressive force, $F$; (**b**) specimen vertical deformation, $d_v$; (**c**) specimen horizontal deformation, $d_h$, with time, $t$, as measured with electrical LVDTs at the center of each specimen lateral face—A, B, C, and D—during the test.

All vertical and horizontal deformations of the specimen returned to zero in the first three cycles of the load after the vertical compressive force *F* was removed. This indicates an elastic behavior of the specimen under a vertical compressive force of $F \leq 530$ kN. With $F > 530$ kN in the last load cycle, the vertical deformation increased non-linearly with force *F*. Such a behavior was especially noticeable on faces C and D of the specimen, which showed the change from an elastic to a plastic deformation regime.

It is worth noting that the electrical LVDT sensors did not detect the onset of cracks in the specimen, because they were installed in the center of the four faces of the specimen, while cracks formed on one of its corners. At the same time, the development of structural cracks led to the breakage of the optical fibers crossing the cracks, and accordingly, to a decrease in the output signals of the monitoring system. In some optical fibers, the breakage was observed visually due to the red light coming from the broken optical fibers (Figure 7).

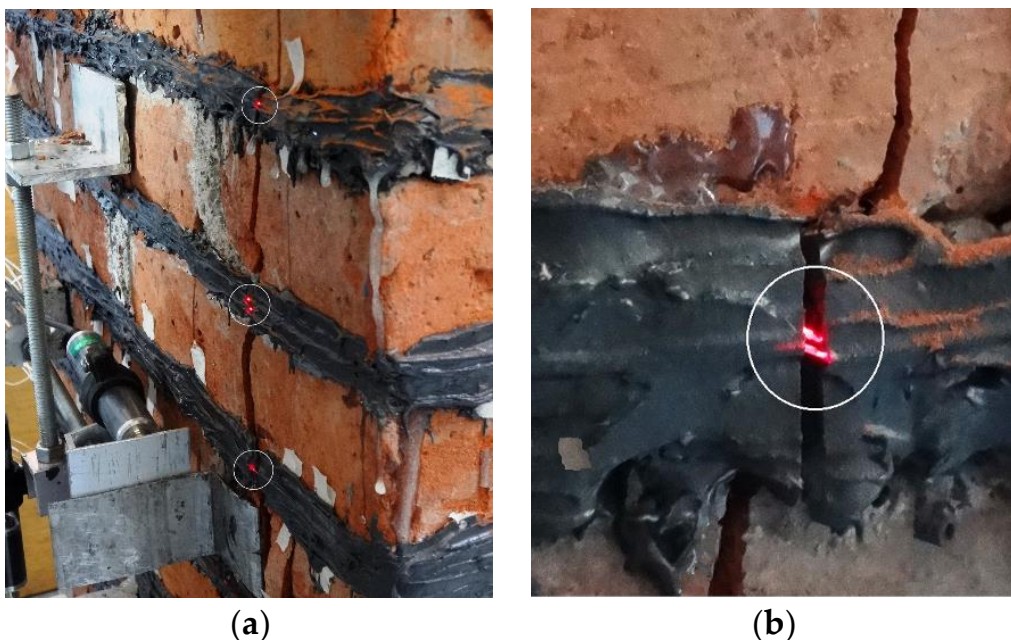

(**a**)            (**b**)

**Figure 7.** Broken optical fibers in locations where they intersect structural cracks: (**a**) completely broken optical fibers; (**b**) optical fibers with broken silica cores and claddings that stay in the intact but stretched elastic coatings.

The plots in Figure 8 show the variation in the applied vertical compressive force on the stack, *F*, and the respective dimensionless amplitude of the output signal, *A*, of the twelve optical fibers with time, *t*. The graphs of the dimensionless amplitude of the signal, *A*, are displayed in groups of four for levels 1 and 2, 3 and 4, and 5 and 6 of the specimen.

The sharp decrease in the signal amplitude observed in these diagrams was caused by optical fiber breaks, complete or partial, due to the occurrence of structural cracks. There are several characteristic points on the plots in Figure 8, denoted with the lowercase italic letters *q, r, s, t,* and *u*. Some optical fibers (1a, 1b, 3a, 3b, 4a, and 4b) broke and consequently decreased their optical transmission at a load of about 400 kN in the fourth load cycle (interval *q-r*), and then they partially recovered it when the load returned to zero (interval *r-s*). Other optical fibers (2a, 2b, 5a, 5b, 6a, and 6b) sharply decreased their optical transmission at a larger load of about 620 kN in the fifth load cycle (point *u*).

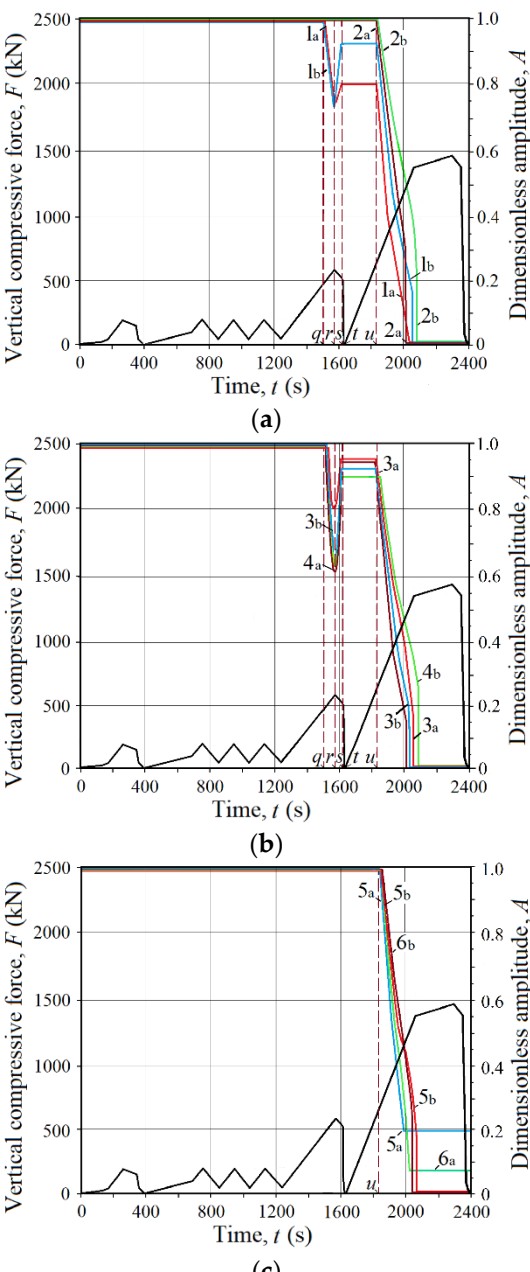

**Figure 8.** Plots of applied vertical compressive force $F$ (—) and dimensionless amplitude of the output signal $A$ (color lines) of the optical fibers mounted at different specimen levels vs. time $t$: (**a**) levels 1 and 2; (**b**) levels 3 and 4; (**c**) levels 5 and 6.

The observed differences in the behavior of some of the optical fibers can be attributed to the structural inhomogeneity of the specimen. This led to inhomogeneous cracking of the specimen under load, and the first structural cracks developed in the most strained parts of the specimen. The optical fibers that crossed the structural cracks were stretched, and their silica members were broken, while the elastic plastic coatings were stretched but not broken. The optical transmission decreased but was not completely interrupted (interval $q$-$r$). With the load dropped to zero, the structural cracks partially closed in the generally elastic specimen. The elastic plastic coatings of the optical fibers contracted and brought the end faces of the broken optical fibers back to a closer position. In this way, optical transmission was partially restored in the said optical fibers (point $t$). In the fifth load cycle, the structural cracks in the specimen opened again and continued to grow. At a load exceeding 620 kN, the optical transmission of all optical fibers dropped sharply (point

*u*). This was caused by the onset and widening of two large vertical structural cracks in the two adjacent lateral faces of the specimen. The width of these cracks varied along the sample height and ranged from 3 mm to 4.6 mm. These cracks caused the breakage of all optical fibers. However, the optical signal of some fibers (5a and 6a) did not vanish completely, as the plastic coatings of these fibers stretched but did not break. As can be seen in Figure 7, the coatings kept both parts of the fibers' broken silica members in a position that provided a finite optical coupling of the two parts and a non-zero amplitude of the optical signal.

### 3.2. Statistical and Probabilistic Evaluation

Statistical analyses yielded a mean structural crack width of $M = 2.68$ mm, a variance of $\sigma^2 = 1.12$ mm$^2$, and a standard deviation of $\sigma = 1.06$ mm ($n = 12$). The relationship between the probability of the optical fiber breakage, $p$, and the width of the structural crack in the test specimen that caused the fiber breakage, $\delta_c$, determined for sequential intervals of a crack width of 0.5 mm is illustrated in Figure 9.

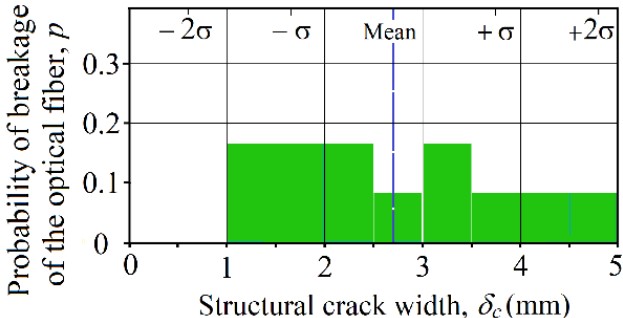

**Figure 9.** Probability of breakage of the optical fiber, $p$, as a function of the width of the structural crack that caused the fiber breakage, $\delta_c$, measured in the test specimen of brick masonry and determined for sequential intervals of a width of 0.5 mm.

As can be seen in the diagram, all optical fibers broke at cracks 1–5 mm wide. Half of all of the optical fibers broke at cracks 1–2.5 mm wide, and three-quarters of all optical fibers broke at cracks 1–3.5 mm wide. Cracks of this width present a medium-to-severe structural hazard, which is targeted in this work.

Furthermore, the probability that at least one fiber of an array of $n$ optical fibers breaks due to a crack with a width in the range $\Delta\delta_c$, $P_1$, is [66]

$$P_1 = 1 - (1 - p)^n \tag{1}$$

where $p$ is the probability of the optical fiber breakage in the range $\Delta\delta_c$ (see Figure 9), and $n$ is the number of optical fibers.

Table 1 shows the probability of breaking at least one fiber in an array of $n$ optical fibers, $P_1$, calculated for some ranges of crack widths $\Delta\delta_c$. Calculations were made for the detection arrays of 1, 2, 4, 6, 8, 10, and 12 optical fibers.

The data in Table 1 show that the probability of the breakage of at least 1 fiber of an array of 12 optical fibers, due to a crack in different parts of the 1–5 mm range, is high ($0.888 < P_1 < 1.0$). A high probability of the breakage of at least one fiber exists even for rather small cracks 1–1.5 mm wide ($P_1 \cong 0.888$). Reducing the size of the detection array even by a factor of two (to 6 fibers) reduces the probability $P_1$, but only the probability of detecting the narrowest cracks degrades significantly ($P_1 = 0.665@\Delta\delta_c = 1$–1.5 mm). The above data illustrate the high probability of detecting cracks with detection arrays of a moderate size, as well as the possibility of trade-off between the size of the detection array and the desired probability of detecting cracks of a certain width.

**Table 1.** Probability that at least one fiber of an array of $n$ optical fibers breaks, $P_1$, calculated for several ranges of crack widths $\Delta\delta_c$ and detection arrays of 1, 2, 4, 6, 8, 10, and 12 optical fibers.

| Number of Optical Fibers of the Detection Array | The Range of Crack Width in Which the Fiber Break Occurs, $\Delta\delta_c$ (mm) | | | | |
|---|---|---|---|---|---|
| $n$ | 1–1.5 ($p = 0.167$) | 1–2 ($p = 0.333$) | 1–3 ($p = 0.583$) | 1–4 ($p = 0.833$) | 1–5 ($p = 1.0$) |
| | Probability that at least one fiber of an array of $n$ optical fibers breaks, $P_1$ | | | | |
| 1 | 0.167 | 0.333 | 0.583 | 0.833 | 1.0 |
| 2 | 0.306 | 0.556 | 0.826 | 0.972 | 1.0 |
| 4 | 0.518 | 0.802 | 0.970 | >0.999 | 1.0 |
| 6 | 0.665 | 0.912 | 0.995 | >0.999 | 1.0 |
| 8 | 0.767 | 0.961 | >0.999 | >0.999 | 1.0 |
| 10 | 0.838 | 0.983 | >0.999 | >0.999 | 1.0 |
| 12 | 0.888 | 0.992 | >0.999 | >0.999 | 1.0 |

## 4. Discussion

Silica-core silica-clad optical fibers can be installed in new and existing buildings and serve as artificial "nerves" that indicate structural failure, just as nerves indicate pain in the body, during the lifetime of the buildings. This technique is simple and inexpensive, for the following reasons:

- The use of ordinary optical fibers of silica without any built-in special sensing elements, such as gratings and tapers. This contributes to the low cost of the optical fiber detector array.
- The use of multi-mode optical fibers rather than single-mode optical fibers. Multi-mode optical fibers have a large core diameter, which allows the use of inexpensive and reliable LEDs instead of expensive and delicate semiconductor lasers needed to efficiently launch light into the small diameter core of single-mode optical fibers.
- The use of optical fibers with their original polymer coating. The removal of the polymer coating used in the manufacture of FBGs with subsequent recoating compromises the strength of the optical fiber. Keeping the original coating intact contributes to the long service life and reliability of optical fibers.
- The use of a sufficiently large number of optical fibers in the crack detection array. This makes it possible to reliably detect structural cracks of a certain width.
- The use of a binary conversion and interpretation of the optical signals.
- The use of time division multiplexing (TDM) to monitor the optical transmission of all fibers of the crack detection array. TDM is much simpler and cheaper than the optical spectral multiplexing used in distributed FBG and OFDR sensors.

Many of the characteristics of this method, as well as the singularities of its application in buildings of different types, require further collaborative study by experts in civil engineering, and measurement and information technologies. It is preferable to install optical fiber arrays at locations of maximum stress and in the direction of the maximum equivalent strain. This ensures the predominantly perpendicular crossing of the optical fibers and the structural cracks. For a given crack, signal loss with a perpendicular crossing is less than that in other cases. More specifically, the signal loss depends on the angle between the crack trace and the fiber axis. Therefore, the expected geometry of the intersection of a fiber with a crack should be taken into account when placing optical fibers on structures. Typical crack patterns in structures under different loadings are generally well known. Moreover, proper stress and strain fields can be obtained by modeling the stress–strain state in the most loaded areas of the structure using numerical methods, such as the finite element method and artificial neural networks [67,68].

There are many options for informing users of the occurrence of structural risk. The simplest form is to notify users about the appearance of a new crack on the monitored building using special displays and sirens at the facility, automatic calls, etc. More advanced and more efficient forms involve the connection of various damage-monitoring modules to data processing facilities by means of different kinds of communication links or networks [69].

## 5. Conclusions

In this work, cracks in a masonry specimen were detected by exploiting the breakage of ordinary silica optical fibers bonded to its surface. For the first time, a reliable detection of structural cracks with a width exceeding a given minimum value was obtained despite the random nature of the ultimate strength of the optical fibers. This was achieved using arrays of several optical fibers placed on the structural element. This paper established a relationship between the number of fibers in the detector array and the probability of detecting a structural crack of a given width. A fairly moderate number of optical fibers (about 6 to 12) was sufficient to detect cracks 1–3 mm wide with a probability of more than 0.99, while the probability of detecting cracks with a width of 1–5 mm tended toward unity.

Overall, this work confirmed the suitability of ordinary silica optical fibers for reliably assessing cracks in masonry that poses a structural risk. The simplicity of this technique and the low cost of ordinary optical fibers make it suitable for broad application in permanent damage-detection systems in buildings in seismic zones. This technique can also potentially be applicable in the detection of structural damage to the load-bearing elements of large land vehicles, aircraft, and ships.

Further work is needed to test this method in various scenarios, to better characterize it, and to obtain more data. In addition, our future objectives are to explore the possibility of detecting narrower and wider structural cracks than those detected in this work.

**Author Contributions:** Conceptualization, methodology, experimentation, and writing—original draft preparation, S.K.; interpretation of the results, data analyses, and validation, V.T. All authors have read and agreed to the published version of the manuscript.

**Funding:** This study was funded by the General Directorate for Academic Staff Affaires (DGAPA), Universidad Nacional Autonoma de Mexico (grant number IT102021).

**Informed Consent Statement:** Not applicable.

**Acknowledgments:** The authors acknowledge Eng. Abraham Roberto Sanchez-Ramirez and Eng. Laura Justina Olivares-Sanchez de Tagle, Institute of Engineering, Universidad Nacional Autonoma de Mexico, for providing structural elements of brick masonry and carrying on the compression tests of the said structural elements, respectively. The authors acknowledge Oleg Kolokoltsev, Institute of Applied Sciences and Technology (ICAT), Universidad Nacional Autonoma de Mexico, Mykola Bobyr, Institute of Mechanical Engineering, National Technical University of Ukraine "Igor Sikorsky Kyiv Polytechnic Institute," Kyiv, Ukraine, and Anastasia K. Lopez, Faculty of Science, University of British Columbia, Vancouver, BC, Canada, for helpful suggestions to the manuscript.

**Conflicts of Interest:** The authors declare no conflict of interest.

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
