# Peer review of "Assessment of Cracking in Masonry Structures Based on the Breakage of Ordinary Silica-Core Silica-Clad Optical Fibers"

_applsci, doi:10.3390/app12146885_

Round 1

Reviewer 1 Report

The paper proposes a crack determination method by  monitoring the interruption of optical fiber transmission signal.The principle and operation of this method are simple.However, I think this method is not applicable in practice.For the whole structure, the distribution of the stree and strain are complex.Besides,At present,a mass of the methods have been proved to obtain the full-field deformation information and structural health monitoring using a few optical fiber sensors.Compared with the existing methods, the proposed scheme is not competitive.

Reviewer 2 Report

This manuscript presents studies on suitability and accuracy of detecting structural cracks in brick masonry by exploiting the breakage of ordinary silica optical fibers placed on its surface. The fibers were bonded to the surface of the masonry specimen with an epoxy adhesive. The specimen was subjected to compressive load test under laboratory conditions. The deformations and cracking of the masonry specimen, and the behavior of the pilot optical signals transmitted through the fibers upon loading of the test specimen were observed. Overall, the conclusions are correct but the reviewer still has some doubts on the novelty of the current work, which should be more clearly presented in the revised version, if a revision is required. Moreover, this article is not well-written. Perhaps, it should be considered as a semi-finished article. It is full of typos and grammatical errors, and sentences need to be further improved.

Comment#1: The part of abstract should be as concise as possible.

Comment#2: There is a clear format error in "(in practice, > 4 GPa, although greatly influenced by presence of flaws and the length of the sample [63]" in line 222.

Comment#3: In the line 235, the authors addressed “The suitability of this optical fiber for the present application was verified by performing a simple tensile test on this optical fiber. The test consisted in bonding an optical fiber sample to two separate ceramic bricks and pushing the two bricks apart until the breakage of the fiber.” Whether the rate of separating the two bricks affects the rate of the change of the optical signal transmitted by the optical fiber. The tensile test with the reference national standard should be listed.

Comment#4: The illustrations in the article are not rigorous enough. The dotted lines in figure 2,6,8 and 9 should be standardized and consistent. The pictures (a), (b) and (c) of Figure 3 and Figure 4 should maintain consistent dimensions. The pictures (a), (b) and (c) of Figure 6 and Figure 8 should maintain consistent dimensions.

Comment#5: In the line 324, the full name of LED "light emitting diodes " should be noted when it first appears in the article, namely, line 231.

Comment#6: In the line 451, the formula appearing in the article should mark the serial number.

Comment#7: In the line 469, "5. Discussion "should be "4. Discussion ", in reviewer’s opinion, this error can be avoided.

Comment#8: Language needs to be improved. Though well structured, a proofreading by native English speakers is still needed to avoid grammatical errors and typos.

Reviewer 3 Report

The abstract is too long and has too much information. The manuscript formatting is not for an article, it is for a book chapter. The breakdown of the first section is very unclear and hard to follow by the reader.

The argument that FBG and OFDR interrogators need "highly qualified personnel to work with systems and interpret measurement results" is speculative and inaccurate. Those interrogators can be simply interfaced to provide more accurate data to any user. In fact, it can be argued that an FBG array or an OFDR interrogator can provide accurate strain measurements of the structure and accurately determine cracking points even without fiber breakage. I'll refer you to "High resolution monitoring of strain fields in concrete during hydraulic fracturing processes".

The breakage of optical fibers will mark the end of their purpose for the proposed setting, while in other settings with FBGs or BOTDR or OFDR, they may still be used, depending on where the breakage occurs on the fiber strand. So, sections 1.4 and 1.5 in the text, and the conclusion that fibers are will break are based on authors logical assumptions/conclusions, but not scientific evidence that was reported in many articles that reported fibers not breaking inside of cracked concrete structures. 

The authors should present their work on detection of fiber breakage and not focus on presenting that fiber breakage is inevitable, which is not the case and of course needs evidence.

The manuscript format is needs adjustments (figures and tables centering, text indentation, .. etc.) 

The labels on the figures are pixelated, the quality needs to be improved. 

There are some parts of text with different colors.

Round 2

Reviewer 1 Report

The author has given a reasonable reply to my question。

Author Response

Manuscript ID: applsci-1702563 Rev 2

Replies to the comments of the 2nd round of Reviewer #1

The authors gratefully thank the Reviewer #1 for his time, consideration, and new comments. Below we present our response to the comments of the 2nd round provided by the 1st Reviewer.

Comment: English language and style are fine/minor spell check required.

Reply: The manuscript has undergone English language editing by MDPI. The text has been checked for correct use of grammar and common technical terms, and edited to a level suitable for reporting research in a scholarly journal. We submit the respective certificate together with the updated manuscript to the Applied Sciences.

Comment: The author has given a reasonable reply to my question。

Reply: Once again, we thank the Reviewer #1 for the time he put in reviewing our manuscript. Since the comments of the Reviewer #1 have been precious, we acknowledge his contribution explicitly.

Reviewer 3 Report

I thank the authors for editing the abstract and conclusions. Unfortunately, the authors have not addressed the reviewer's concerns in their revised manuscript. I've only mentioned the Chen article as an example of fibers working and not breaking, I did not ask for it to be included. 

I encourage the authors to address the issues raised in the reviews in their manuscript.

Author Response

Manuscript ID: applsci-1702563 Rev 2

Replies to the comments of the 2nd round of Reviewer #3

We gratefully thank the Reviewer for his time, consideration, and new comments. Below we present our response to the comments of the 2nd round provided by the 3rd Reviewer.

Comment: English language and style are fine/minor spell check required.

Reply: The manuscript has undergone English language editing by MDPI. The text has been checked for correct use of grammar and common technical terms and edited to a level suitable for reporting research in a scholarly journal. We submit the respective certificate together with the updated manuscript to the Applied Sciences.

Comment: I thank the authors for editing the abstract and conclusions.

Reply: We thank the Reviewer #3 for the respective suggestion since he urged us to rewrite the abstract and conclusions and thus contributed to improving the quality and readability of the manuscript.

Comment: Unfortunately, the authors have not addressed the reviewer's concerns in their revised manuscript.

Reply: We provided the 3rd Reviewer with point-to-point replies to all his comments and concerns of the 1st round of revisions and fulfilled all suggestions that the 3rd Reviewer made in a specific and clear form.

     At the same time, we are not sure whether we fully understand these comments: “The manuscript formatting is not for an article, it is for a book chapter. The breakdown of the first section is very unclear and hard to follow by the reader”.

      The authors followed the MDPI rules for authors and its template for writing, defining the manuscript content and length, dividing section in subsections and formatting the manuscript. Following the comment of the 3rd reviewer, we edited the lines 38 and 46-52 to make the Introduction section more readable.

      In what refers to the “book chapter formatting”, the book chapters are usually just more detailed and longer than journal articles. Nevertheless, we take into account that the majority of readers of journal articles are PhD students, postgraduates, postdocs, and other types of academics who may be more or less knowledgeable about the topic addressed in the paper (https://doi.org/10.1002/asi.23286). Therefore, we believe that the detailed and accessible writing style is beneficial to the audience.

      The authors would be more than happy to make any further specific changes that will improve the manuscript.

Comment: I've only mentioned the Chen article as an example of fibers working and not breaking, I did not ask for it to be included. 

Reply: We thank the reviewer for pointing to the publication in question and consider it useful to cite this work in our manuscript.

Comment: I encourage the authors to address the issues raised in the reviews in their manuscript.

Reply: We would appreciate it if you could point out specific sentences and/or paragraphs for improvement in the next round of revisions.

Once again, thank the 3rd Reviewer for his time, consideration, and comments.

Regards,

SK

Corresponding coauthor
